# Muon decay

**W. Fetscher⋆**

Institute for Physics and Astrophysics, ETH Zürich

⋆ fetscher@phys.ethz.ch

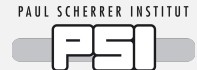
## Abstract

The decay of the muon has been studied at PSI with several precision measurements: The longitudinal polarization $P_{\mathrm{L}}(E)$ with the muon decay parameters $\xi'$, $\xi''$, the Time-Reversal Invariance (TRI) conserving transverse polarization $P_{\mathrm{T_1}}(E)$ with the muon decay parameters $\eta$, $\eta''$, the TRI violating transverse polarization $P_{\mathrm{T_2}}(E)$, with $\alpha'/A$, $\beta'/A$ and the muon decay asymmetry with $P_\mu\xi$. The detailed theoretical analysis of all measurements of normal and inverse muon decay has led for the first time to a lower limit $|g_{LL}^V| > 0.960$ ("$V-A$") and upper limits for nine other possible complex couplings, especially the scalar coupling $|g_{LL}^S| < 0.550$ which had not been excluded before.

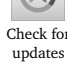
## 6.1 Introduction

Muon decay, $\mu^+ \rightarrow \overline{\nu}_\mu e^+ \nu_e$, as a purely leptonic process, provides a precise source of information on the charged current weak interaction. Before the advent of the meson factories LAMPF, TRIUMF and SIN, experimental results were scarce and theoretical descriptions inappropriate to uniquely deduce the interaction. In a combined effort, the ETH-SIN group has performed decisive precision measurements and, simultaneously, developed the theoretical description in a way that allowed the determination of the interaction from experimental results, taken exclusively from normal and inverse muon decay ($\nu_\mu + e^- \rightarrow \mu^- + \nu_e$).

## 6.2 General Matrix Element

The three leptonic decays $\mu^+ \rightarrow \bar{\nu}_\mu e^+ \nu_e$, $\tau^+ \rightarrow \bar{\nu}_\tau \mu^+ \nu_\mu$ and $\tau^+ \rightarrow \bar{\nu}_\tau e^+ \nu_e$, as well as their charge conjugate decays, can be described by the most general, local, derivative-free and lepton-number conserving four-fermion contact interaction Hamiltonian. The contact interaction allows the use of equivalent Hamiltonians, which differ in the way the fermions are grouped together [1, 2]. The older literature preferred a "charge retention" form with parity-odd and parity-even terms in which $e^+$ and $\mu^+$, as the usually detected particles, were grouped

together [3, 4]. This had the advantage that limits to some coupling constants could be obtained from then existing results. The disadvantage was that this Hamiltonian represents interactions proceeding via the exchange of a neutral boson $X$ that would carry the lepton numbers both of muon and electron, and so would not be universal. The use of a "charge-changing" form, where the charged leptons are grouped with their neutrinos and which is adapted to charged boson exchange, results in absolute values of differences of coupling constants. Both of these forms are complicated by the fact that a fully parity-violating interaction, such as e.g. the $V - A$- interaction, is represented by *four* coupling constants $C_V$, $C'_V$, $C_A$ and $C'_A$.

In the following, we will use a charge-changing Hamiltonian characterized by fields of definite chirality [5, 6]. We use the notation of Fetscher *et. al.* [7], which in turn uses the sign conventions and definitions of Scheck [8]. The general matrix element can then be written as

$$M = 4 \frac{G_F}{\sqrt{2}} \sum_{\substack{\gamma = S, V, T \\ \varepsilon, \mu = R, L}} g_{\varepsilon\mu}^{\gamma} \langle \bar{e}_\varepsilon | \Gamma^\gamma | (\nu_e)_n \rangle \langle (\bar{\nu}_\mu)_m | \Gamma_\gamma | \mu_\mu \rangle. \tag{6.1}$$

Here, $G_F$ is the Fermi coupling constant, while $\gamma = S, V, T$ indicates a 4-scalar, 4-vector, or 4-tensor interaction; the corresponding $\Gamma^\gamma$ could be either Dirac $\gamma$ matrices or, when using the Weyl spinors of Eqs. (6.2) to (6.4), Pauli matrices. The indices $\varepsilon, \mu = R, L$ indicate the chirality (right- or left-handed) of the spinors of the electron or muon. The chiralities $n$ and $m$ of the $\nu_e$ and $\bar{\nu}_\mu$ are then determined by the values of $\gamma, \varepsilon$, and $\mu$. In this picture, the coupling constants $g_{\varepsilon\mu}^\gamma$ have a simple physical interpretation: $n_\gamma |g_{\varepsilon\mu}^\gamma|^2$ is equal to the (relative) probability for a $\mu$-handed muon to decay into an $\varepsilon$-handed electron by the interaction $\Gamma^\gamma$; the factors $n_S = 1/4$, $n_V = 1$ and $n_T = 3$ take care of the proper normalisation. The standard model thus corresponds to $g_{LL}^V = 1$, with all other couplings being zero.

We emphasise that here right- and left-handed definitely means chirality and not helicity. The left-handed spinor $\overset{\circ}{\chi}$ of a fermion in its rest system transforms under a Lorentz-boost as

$$\chi_L(\boldsymbol{p}) = \frac{(E+m)\sigma^0 - \boldsymbol{p} \cdot \boldsymbol{\sigma}}{\sqrt{2m(E+m)}} \overset{\circ}{\chi}, \tag{6.2}$$

where $\sigma^0$ and $\boldsymbol{\sigma}$ are the four Pauli matrices. By a parity operation, $\chi_L(\boldsymbol{p})$ becomes the right-handed spinor $\chi_R(\boldsymbol{p})$. Left- and right-handed spinors are contained in separate $\mathbb{C}$ 2-spaces. The right-handed spinor transforms under a Lorentz-boost as

$$\chi_R(\boldsymbol{p}) = \frac{(E+m)\sigma^0 + \boldsymbol{p} \cdot \boldsymbol{\sigma}}{\sqrt{2m(E+m)}} \overset{\circ}{\chi}. \tag{6.3}$$

The spinor of the antiparticle is given by

$$\varphi_L(\boldsymbol{p}) = +i\sigma^2 \chi_R^*(\boldsymbol{p}) \quad \text{and} \quad \varphi_R(\boldsymbol{p}) = -i\sigma^2 \chi_L^*(\boldsymbol{p}). \tag{6.4}$$

## 6.3 Observables

The differential decay probability to obtain an $e^\pm$ with (reduced) energy between $x$ and $x + \mathrm{d}x$, emitted in the direction $\hat{\boldsymbol{x}}_3$ at an angle between $\vartheta$ and $\vartheta + \mathrm{d}\vartheta$ with respect to the muon polarization vector $\boldsymbol{P}_\mu$, and with its spin parallel to the arbitrary direction $\hat{\boldsymbol{\zeta}}$, neglecting radiative corrections, is given by

$$\frac{\mathrm{d}^2\Gamma}{\mathrm{d}x \, \mathrm{d}\cos\vartheta} = \frac{m_\mu}{4\pi^3} W_{e\mu}^4 G_F^2 \sqrt{x^2 - x_0^2} \cdot \left\{ F_{\mathrm{IS}}(x) \pm P_\mu \cos\vartheta F_{\mathrm{AS}}(x) \right\} \cdot \left\{ 1 + \hat{\boldsymbol{\zeta}} \cdot \boldsymbol{P}_e(x, \vartheta) \right\}. \tag{6.5}$$

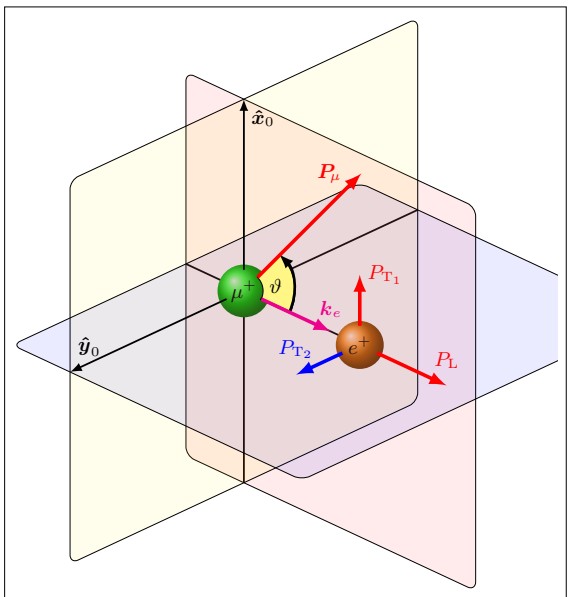

Figure 6.1: Definition of the observables in polarized muon decay: muon polarization $\boldsymbol{P}_\mu$, positron momentum $\boldsymbol{k}_e$, longitudinal positron polarization $P_{\mathrm{L}}$, transverse positron polarization $(P_{\mathrm{T}_1}, P_{\mathrm{T}_2})$ and angle of emission $\vartheta$ (relative to $\boldsymbol{P}_\mu$). Time reversal invariance is violated if $P_{\mathrm{T}_2} \neq 0$. From [11].

Here, $W_{e\mu} = \max(E_e) = (m_\mu^2 + m_e^2)/(2m_\mu)$ is the maximum $e^\pm$ energy, $x = E_e/W_{e\mu}$ is the reduced energy, $x_0 = m_e/W_{e\mu} = 9.67 \times 10^{-3}$, and $P_\mu = |\boldsymbol{P}_\mu|$ is the degree of muon polarization. $\hat{\boldsymbol{\zeta}}$ is the direction in which a perfect polarization-sensitive electron detector is most sensitive. The isotropic part of the spectrum, $F_{\mathrm{IS}}(x)$, the anisotropic part $F_{\mathrm{AS}}(x)$, and the electron polarization, $\boldsymbol{P}_e(x,\vartheta)$, may be parameterized by the Michel parameter $\rho$ [1], by $\eta$ [9], by $\xi$ and $\delta$ [3, 10], *etc.* These are bilinear combinations of the coupling constants $g_{\varepsilon\mu}^\gamma$, which occur in the matrix element (given below).

If the masses of the neutrinos as well as $x_0$ are neglected, the energy and angular distribution of the electron in the rest frame of a muon ($\mu^\pm$) measured by a polarization insensitive detector is given by

$$\frac{\mathrm{d}^2\Gamma}{\mathrm{d}x\,\mathrm{d}\cos\vartheta} \sim x^2 \cdot \left\{ 3(1-x) + \frac{2\rho}{3}(4x-3) + 3\eta x_0(1-x)/x \right.$$
$$\left. \pm P_\mu \cdot \xi \cdot \cos\vartheta \left[ 1 - x + \frac{2\delta}{3}(4x-3) \right] \right\}. \tag{6.6}$$

Here, $\vartheta$ is the angle between the electron momentum and the muon spin, and $x \equiv 2E_e/m_\mu$. Within the Standard Model, we obtain $\rho = \xi\delta = 3/4$, $\xi = 1$, $\eta = 0$ and the differential decay rate is

$$\frac{\mathrm{d}^2\Gamma}{\mathrm{d}x\,\mathrm{d}\cos\vartheta} = \frac{G_F^2 m_\mu^5}{192\pi^3} \left[ 3 - 2x \pm P_\mu \cos\vartheta(2x-1) \right] x^2. \tag{6.7}$$

The coefficient in front of the square bracket is the total decay rate.

The observables in the decay of polarized muons are shown in Figure 6.1. We have defined a right-handed coordinate system with

$$\hat{\boldsymbol{z}}_0 = \frac{\boldsymbol{k}_e}{|\boldsymbol{k}_e|}, \quad \hat{\boldsymbol{y}}_0 = \frac{\boldsymbol{k}_e \times \boldsymbol{P}_\mu}{|\boldsymbol{k}_e \times \boldsymbol{P}_\mu|}, \quad \hat{\boldsymbol{x}}_0 = \hat{\boldsymbol{y}}_0 \times \hat{\boldsymbol{z}}_0. \tag{6.8}$$

Here, $\boldsymbol{k}_e$ is the momentum vector of the electron, while $P_{\mathrm{L}}$ designates the longitudinal polarization, $P_{\mathrm{T}_1}$ the transverse component of $\boldsymbol{P}_e$ lying in the plane defined by $\boldsymbol{k}_e$ and $\boldsymbol{P}_\mu$, and $P_{\mathrm{T}_2}$ is the component perpendicular to that plane. $P_{\mathrm{T}_2} \neq 0$ signals violation of time-reversal symmetry. These polarization components are

$$P_{\mathrm{T}_1}(x,\vartheta) = \frac{P_\mu \sin\vartheta \cdot F_{\mathrm{T}_1}(x)}{F_{\mathrm{IS}}(x) \pm P_\mu \cos\vartheta \cdot F_{\mathrm{AS}}(x)}, \tag{6.9}$$

$$P_{\mathrm{T}_2}(x,\vartheta) = \frac{P_\mu \sin\vartheta \cdot F_{\mathrm{T}_2}(x)}{F_{\mathrm{IS}}(x) \pm P_\mu \cos\vartheta \cdot F_{\mathrm{AS}}(x)}, \tag{6.10}$$

$$P_{\mathrm{L}}(x,\vartheta) = \frac{\pm F_{\mathrm{IP}}(x) + P_\mu \cos\vartheta \cdot F_{\mathrm{AP}}(x)}{F_{\mathrm{IS}}(x) \pm P_\mu \cos\vartheta \cdot F_{\mathrm{AS}}(x)}. \tag{6.11}$$

If only the neutrino masses are neglected, and if the $e^\pm$ polarization is detected, then the functions in (6.5) can be decomposed as [12]

$$F_\nu(x) = F_\nu^{V-A}(x) + G_\nu(x), \tag{6.12}$$

where $G_\nu(x) \equiv 0$ for $g_{LL}^V = 1$ ("$V-A$"). Physics beyond the Standard Model would thus be contained *exclusively* in the $G_\nu(x)$. The index $\nu$ stands for IS (isotropic part of the spectrum), AS (anisotropic part of the spectrum), $\mathrm{T}_1$ (transverse polarization $P_{\mathrm{T}_1}$), $\mathrm{T}_2$ (transverse polarization $P_{\mathrm{T}_2}$), IP (isotropic part of the longitudinal polarization) and AP (anisotropic part of the longitudinal polarization). The $F_\nu^{V-A}(x)$ do not depend on specific decay parameters:

$$F_{\mathrm{IS}}^{V-A}(x) = \tfrac{1}{6}\left\{-2x^2 + 3x - x_0^2\right\}, \tag{6.13a}$$

$$F_{\mathrm{AS}}^{V-A}(x) = \tfrac{1}{6}\left(x^2 - x_0^2\right)^{1/2}\left\{2x - 2 + \left(1 - x_0^2\right)^{1/2}\right\}, \tag{6.13b}$$

$$F_{\mathrm{T}_1}^{V-A}(x) = -\tfrac{1}{6}(1-x)x_0, \tag{6.13c}$$

$$F_{\mathrm{T}_2}^{V-A}(x) = 0, \tag{6.13d}$$

$$F_{\mathrm{IP}}^{V-A}(x) = \tfrac{1}{6}\left(x^2 - x_0^2\right)^{1/2}\left\{-2x + 2 + \left(1 - x_0^2\right)^{1/2}\right\}, \tag{6.13e}$$

$$F_{\mathrm{AP}}^{V-A}(x) = \tfrac{1}{6}\left\{2x^2 - x - x_0^2\right\}. \tag{6.13f}$$

The functions $G_\nu(x)$ depend on the decay parameters $\rho, \xi'', \xi', \xi, \delta, \eta, \eta'', \alpha'/A, \beta'/A$, where $\eta = (\alpha - 2\beta)/A$ and $\eta'' = (3\alpha + 2\beta)/A$:

$$G_{\mathrm{IS}}(x) = \tfrac{1}{9}\left\{2\left(\rho - \tfrac{3}{4}\right)\left(4x^2 - 3x - x_0^2\right) + 9\eta(1-x)x_0\right\}, \tag{6.14a}$$

$$\begin{aligned} G_{\mathrm{AS}}(x) = \tfrac{1}{9}\left(x^2 - x_0^2\right)^{1/2}\Big\{ & 3(\xi-1)(1-x), \\ & +2\left(\xi\delta - \tfrac{3}{4}\right)\left(4x - 4 + \left(1 - x_0^2\right)^{1/2}\right)\Big\}, \end{aligned} \tag{6.14b}$$

$$\begin{aligned} G_{\mathrm{T}_1}(x) = \tfrac{1}{12}\Big\{ & -2\left[(\xi''-1) + 12\left(\rho - \tfrac{3}{4}\right)\right](1-x)x_0 \\ & -3\eta\left(x^2 - x_0^2\right) + \eta''\left(-3x^2 + 4x - x_0^2\right)\Big\}, \end{aligned} \tag{6.14c}$$

$$G_{\mathrm{T}_2}(x) = \tfrac{1}{3}\left(x^2 - x_0^2\right)^{1/2}\left\{3\frac{\alpha'}{A}(1-x) + 2\frac{\beta'}{A}\left(1 - x_0^2\right)^{1/2}\right\}, \tag{6.14d}$$

$$\begin{aligned} G_{\mathrm{IP}}(x) = \tfrac{1}{54}\left(x^2 - x_0^2\right)^{1/2}\Big\{ & 9(\xi'-1)\left[-2x + 2 + \left(1 - x_0^2\right)^{1/2}\right] \\ & +4\xi\left(\delta - \tfrac{3}{4}\right)\left[4x - 4 + \left(1 - x_0^2\right)^{1/2}\right]\Big\}, \end{aligned} \tag{6.14e}$$

$$\begin{aligned} G_{\mathrm{AP}}(x) = \tfrac{1}{6}\Big\{ & (\xi''-1)\left(2a^2 - x - x_0^2\right) + 4\left(\rho - \tfrac{3}{4}\right)\left(4x^2 - 3x - x_0^2\right) \\ & +2\eta''(1-x)x_0\Big\}. \end{aligned} \tag{6.14f}$$

Several of the decay parameters $\{\rho, \xi, \xi', \xi'', \delta, \eta, \eta'', \alpha/A, \beta/A, \alpha'/A, \beta'/A\}$, which are not all independent, have been measured in the past. Past experiments have also been analyzed using the parameters $a$, $b$, $c$, $a'$, $b'$, $c'$, $\alpha/A$, $\beta/A$, $\alpha'/A$, $\beta'/A$ (and $\eta = (\alpha - 2\beta)/2A$), as defined by Kinoshita and Sirlin [3, 10]. They serve as a model-independent summary of all possible measurements on the decay electron (see Listings below). The relations between the two sets of parameters are

$$\rho - \tfrac{3}{4} = \tfrac{3}{4}(-a + 2c)/A, \tag{6.15}$$

$$\eta = (\alpha - 2\beta)/A, \tag{6.16}$$

$$\eta'' = (3\alpha + 2\beta)/A, \tag{6.17}$$

$$\delta - \tfrac{3}{4} = \tfrac{9}{4}\frac{(a' - 2c')/A}{1 - [a + 3a' + 4(b + b') + 6c - 14c']/A}, \tag{6.18}$$

$$1 - \xi\frac{\delta}{\rho} = 4\frac{[(b + b') + 2(c - c')]/A}{1 - (a - 2c)/A}, \tag{6.19}$$

$$1 - \xi' = [(a + a') + 4(b + b') + 6(c + c')]/A, \tag{6.20}$$

$$1 - \xi'' = (-2a + 20c)/A, \tag{6.21}$$

where

$$A = a + 4b + 6c. \tag{6.22}$$

The ten complex amplitudes $g^{\gamma}_{\varepsilon\mu}$ ($g^T_{RR}$ and $g^T_{LL}$ are identically zero) and $G_F$ constitute 20 independent (real) parameters to be determined by experiment. The Standard Model interaction corresponds to one single amplitude $g^V_{LL}$ being unity and all the others being zero.

## 6.4 Lorentz Structure

The nine parameters $\{\rho, \xi, \xi', \xi'', \delta, \eta, \eta'', \alpha'/A, \beta'/A\}$ describing the electron spectrum, decay asymmetry and polarization vector can be represented [3] by the intermediate quantities $\{a, a', \alpha, \alpha', b, b', \beta, \beta', c, c'\}$, whose values are known from experiment [13]. They are all real, bilinear combinations of the coupling constants:

$$a = 16\left(|g^V_{RL}|^2 + |g^V_{LR}|^2\right) + |g^S_{RL} + 6g^T_{RL}|^2 + |g^S_{LR} + 6g^T_{LR}|^2, \tag{6.23a}$$

$$a' = 16\left(|g^V_{RL}|^2 - |g^V_{LR}|^2\right) + |g^S_{RL} + 6g^T_{RL}|^2 - |g^S_{LR} + 6g^T_{LR}|^2, \tag{6.23b}$$

$$\alpha = 8Re\left\{g^V_{LR}(g^{S*}_{RL} + 6g^{T*}_{RL}) + g^V_{RL}(g^{S*}_{LR} + 6g^{T*}_{LR})\right\}, \tag{6.23c}$$

$$\alpha' = 8Im\left\{g^V_{LR}(g^{S*}_{RL} + 6g^{T*}_{RL}) - g^V_{RL}(g^{S*}_{LR} + 6g^{T*}_{LR})\right\}, \tag{6.23d}$$

$$b = 4\left(|g^V_{RR}|^2 + |g^V_{LL}|^2\right) + |g^S_{RR}|^2 + |g^S_{LL}|^2, \tag{6.23e}$$

$$b' = 4\left(|g^V_{RR}|^2 - |g^V_{LL}|^2\right) + |g^S_{RR}|^2 - |g^S_{LL}|^2, \tag{6.23f}$$

$$\beta = -4Re\{g^V_{RR}g^{S*}_{LL} + g^V_{LL}g^{S*}_{RR}\}, \tag{6.23g}$$

$$\beta' = 4Im\{g^V_{RR}g^{S*}_{LL} - g^V_{LL}g^{S*}_{RR}\}, \tag{6.23h}$$

$$c = \tfrac{1}{2}\{|g^S_{RL} - 2g^T_{RL}|^2 + |g^S_{LR} - 2g^T_{LR}|^2\}, \tag{6.23i}$$

$$c' = \tfrac{1}{2}\{|g^S_{RL} - 2g^T_{RL}|^2 - |g^S_{LR} - 2g^T_{LR}|^2\}. \tag{6.23j}$$

From (6.23a) to (6.23j) it can be seen that these quantities are not completely independent. The transformation from the 20-dimensional space of the complex $g^{\gamma}_{\varepsilon\mu}$ to the 10-dimensional

space of the $\{a,\ldots,c'\}$ leads to the following constraints [14]:

$$a \geq 0 \qquad\qquad a^2 \geq a'^2 + \alpha^2 + \alpha'^2\,, \tag{6.24}$$

$$b \geq 0 \qquad\qquad b^2 \geq b'^2 + \beta^2 + \beta'^2\,, \tag{6.25}$$

$$c \geq 0 \qquad\qquad c^2 \geq c'^2\,. \tag{6.26}$$

These constraints are very important for any general analysis of muon decay, as they strongly influence the final errors of the quantities they relate.

The precise measurement of individual decay parameters alone generally does not give conclusive information about the kind of interaction due to the many different couplings and the interference terms between them. A good example for this is the famous Michel parameter $\varrho$. A precise measurement yielding the value $3/4$ as predicted by $V-A$ by no means establishes the $V-A$ interaction. In fact any interaction consisting of an arbitrary combination of $g_{LL}^S$, $g_{LR}^S$, $g_{RL}^S$, $g_{RR}^S$, $g_{RR}^V$ and $g_{LL}^V$ will yield exactly $\varrho = \frac{3}{4}$. This can be seen if we write $\varrho$ in the form [15]

$$\varrho - \tfrac{3}{4} = -\tfrac{3}{4}\{|g_{LR}^V|^2 + |g_{RL}^V|^2 + 2(|g_{LR}^T|^2 + |g_{RL}^T|^2) - Re(g_{LR}^S g_{LR}^{T*} + g_{RL}^S g_{RL}^{T*})\}\,. \tag{6.27}$$

For $\varrho = 3/4$ and $g_{LR}^T = g_{RL}^T = 0$ (no tensor interaction) we find $g_{LR}^V = g_{RL}^V = 0$, with all the remaining six couplings being arbitrary!

The magnitude of the interaction is contained in the Fermi coupling constant $G_{\mathrm{F}}$. Thus the $g_{\mu\nu}^\gamma$ may be normalized, dimensionless coupling constants, resulting in

$$A \equiv a + 4b + 6c = 16\,. \tag{6.28}$$

This is equivalent to

$$Q_{RR} + Q_{LR} + Q_{RL} + Q_{LL} = 1\,, \tag{6.29}$$

where

$$Q_{RR} = \tfrac{1}{4}|g_{RR}^S|^2 + |g_{RR}^V|^2\,, \tag{6.30}$$

$$Q_{RL} = \tfrac{1}{4}|g_{RL}^S|^2 + |g_{RL}^V|^2 + 3|g_{RL}^T|^2\,, \tag{6.31}$$

$$Q_{LR} = \tfrac{1}{4}|g_{LR}^S|^2 + |g_{LR}^V|^2 + 3|g_{LR}^T|^2\,, \tag{6.32}$$

$$Q_{LL} = \tfrac{1}{4}|g_{LL}^S|^2 + |g_{LL}^V|^2\,. \tag{6.33}$$

We note that $0 \leq Q_{\varepsilon\mu} \leq 1$ and $\sum_{\varepsilon\mu} Q_{\varepsilon\mu} = 1$. $Q_{\varepsilon\mu}$ is then the probability for the decay of a muon of handedness $\mu$ into an electron of handedness $\varepsilon$. The main point is now that the $Q_{\varepsilon\mu}$ can be expressed by the known quantities $\{a,\ldots,c'\}$ [7]:

$$Q_{RR} = 2(b + b')/A\,, \tag{6.34}$$

$$Q_{RL} = [(a - a') + 6(c - c')]/(2A)\,, \tag{6.35}$$

$$Q_{LR} = [(a + a') + 6(c + c')]/(2A)\,, \tag{6.36}$$

$$Q_{LL} = 2(b - b')/A\,. \tag{6.37}$$

In the Standard Model, $Q_{LL} = 1$ while the others are zero. The existing measurements show that the three probabilities $Q_{RR}, Q_{LR}$ and $Q_{LL}$ are zero, within errors. This gives upper limits to the absolute values of eight of the ten complex coupling constants. Furthermore, we find that $Q_{LL}$ is bounded by a lower limit which shows that both muon and electron are left-handed. It can be seen from (6.33), however, that the data from the measurements of the muon and the electron observables do not allow one to distinguish a vector ($g_{LL}^V$) from a scalar ($g_{LL}^S$)

interaction. This type of ambiguity has been noted before in the context of a different Hamiltonian [16, 17] and electron-neutrino correlation measurements (not performed up to date) have been proposed. The total rate $S$, normalized to the rate predicted by $V - A$ for the reaction $\nu_\mu + e^- \rightarrow \mu^- + \nu_e$ with $\nu_\mu$ of negative helicity, has been found to be close to 1 [17, 18]. $S$ effectively depends only on those five coupling constants $g^V_{LL}$, $g^V_{RL}$, $g^S_{LR}$, $g^T_{LR}$ and $g^S_{RR}$ that describe interactions with a left-handed $\nu_\mu$. The four latter coupling constants are found to be small. One thus obtains [7]

$$S = |g^V_{LL}|^2, \tag{6.38}$$

which yields a *lower* limit for $|g^V_{LL}|$, and through the normalisation requirement (6.29) an upper limit for the remaining $|g^S_{LL}|$:

$$|g^S_{LL}| < 2\sqrt{1-S}. \tag{6.39}$$

Thus the weak interaction has been completely determined for muon decay using only data from this purely leptonic interaction.

## 6.5 Experiments

### 6.5.1 Longitudinal Positron Polarization

The measurement of the longitudinal polarization $P_L$ of the electrons from the decay of polarized or unpolarized muons allows the determination of the parameters $\xi'$ and $\xi''$, as can be seen from Eqs. (6.11), (6.12), (6.14e) and (6.14f). The parameter $\xi'$ is of special interest. In terms of the coupling constants $g^\gamma_{\varepsilon\mu}$ we have

$$\begin{aligned} 1 - \xi' &= \tfrac{1}{2}\left\{ 4 \cdot \left( |g^V_{RR}|^2 + |g^V_{RL}|^2 \right) + \left( |g^S_{RR}|^2 + |g^S_{RL}|^2 \right) + 12 \cdot |g^T_{RL}|^2 \right\} \\ &= 2(Q_{RR} + Q_{RL}) \equiv 2Q^e_R, \end{aligned} \tag{6.40}$$

where $Q^e_R$ is the probability of the decay of a muon with chirality $\mu$ into an electron with chirality $\varepsilon$. Note that (6.40) is a sum of absolute squares where only coupling constants with $\varepsilon = R$ appear. A deviation of $\xi'$ from 1 would require the existence of a coupling with the right-handed components of the electron, i.e. at least one $g^\gamma_{R\mu} \neq 0$. Conversely, a measurement with the result $\xi' = 1$ would prove that the coupling acts exclusively on the left-handed component of the electron.

To determine $\xi'$, the longitudinal polarization $P_L$ of the electrons from unpolarized muons has been measured. For the purpose of illustration, we neglect the electron mass $m_e$ and use the experimentally well confirmed values $\varrho = \delta = \tfrac{3}{4}$ and obtain from (6.11)

$$\xi' = P_L. \tag{6.41}$$

The measurement of the electron's longitudinal polarization $P_L$ consists of a comparison with the spin polarization of the electrons contained in a piece of saturated ferromagnetic material [19–21]. The comparison is done by scattering the decay electrons from the electrons of a ferromagnet, using the fact that relativistic electron-electron scattering most often occurs when the two spins have opposite directions.

The experiment was performed at the $\pi$E1 beam line at SIN. A schematic view of the apparatus is shown in Figure 6.2. The 150-MeV/$c$ $\pi^+$ beam was stopped in an oak target, where the $\pi^+$ decay resulted in an unpolarized sample of $\mu^+$ within the oak target. Positrons from muon decay crossed a magnetised iron foil, where they could annihilate in flight with polarized electrons (ANN), $e^+e^- \rightarrow \gamma\gamma$, or scatter elastically: Bhabha-scattering BHA), $e^+e^- \rightarrow e^+e^-$. Both

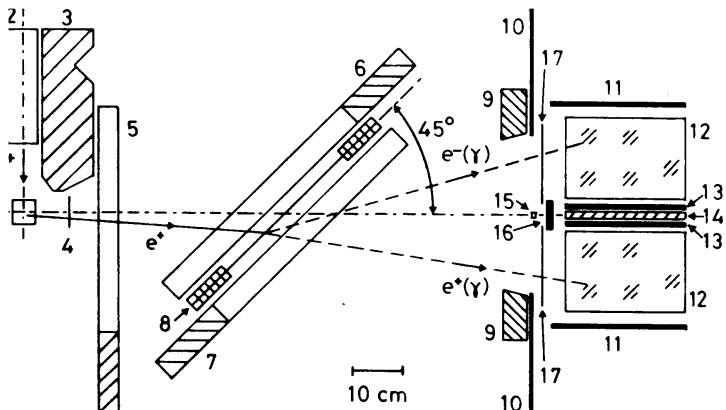

Figure 6.2: Schematic top view of the apparatus used for the measurement of $P_L$. A typical event is shown for either ANN or BHA. The experimental arrangement: (1) oak stopping target; (2) Be-$CH_2$ moderator; (3) shielding; (4) timing counter; (5), (6), and (7) multiwire proportional chambers labeled in the text $WC_1$, $WC_2$, and $WC_3$, respectively); and (8) magnet with iron foil. The total-absorption spectrometer is symmetric to the central axis. It consists of (12) four NaI detectors (only the upper pair is shown); (9) square Pb collimator; (10) square-aperture anticoincidence counter; (15) Am-Be calibration source; (17) four electron-identification counters; (16) vertical anticoincidence counter and monitor; (11) and (13) vertical anticoincidence counters; 14) vertical Fe-Pb photon converters. Not shown are the horizontal counterparts of (11), (13), (14) and (16).

reactions have high analysing powers up to 90 %. The electron polarization in the iron foil was $(54.44 \pm 0.56) \times 10^{-3}$. The final result of this experiment is [14]

$$\langle |P_L| \rangle = 0.998 \pm 0.042 \,. \tag{6.42}$$

From the resulting error of $\xi'$, which is dominated by the error of $\langle |P_L| \rangle$, upper limits for all couplings of right-handed electrons to muons (of any handedness) $g_{R\mu}^\gamma$, $\mu = R, L$, follow, in principle, from (6.40). Improved values of these limits are obtained for $|g_{RL}^V|$ and $|g_{RL}^S + 6g_{RL}^T|$ by also considering

$$B_{RL} = \tfrac{1}{16}|g_{RL}^S + 6g_{RL}^T|^2 + |g_{RL}^V|^2 = \tfrac{1}{2A}(a + a') \,. \tag{6.43}$$

The parameter $\xi''$ in $\mu^+$ decay has been determined from a measurement of $P_L(x, \vartheta)$ as a function of the reduced energy $x$ and the angle $\vartheta$ between the muon spin and the positron momentum [14]. The precision of the measured combination $(\xi'' - \xi\xi')/\xi = -0.35 \pm 0.33$ does, however, not lead to better constraints of the couplings. With a new dedicated setup this value was considerably improved to [22]

$$\xi'' = 0.981 \pm 0.045_{\text{stat.}} \pm 0.003_{\text{syst.}} \,. \tag{6.44}$$

### 6.5.2 Transverse Positron Polarization

The transverse electron polarization $\boldsymbol{P}_T = (P_{T_1}, P_{T_2})$ is defined in Figure 6.1 and Eqs. (6.9) and (6.10). Independent of any assumption about the mechanism of muon decay or even

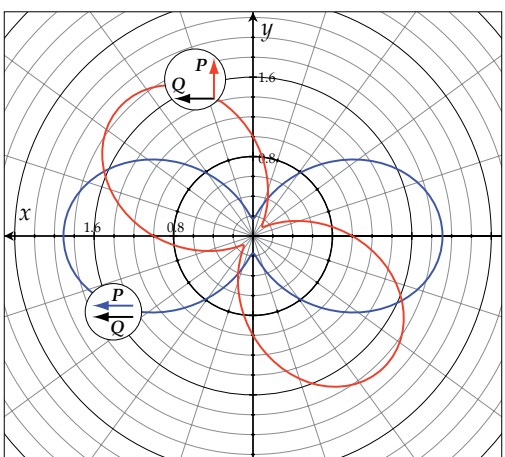

Figure 6.3: Intensity distributions of the annihilation photons at $E_3 = E_4 = 50\,m_e$ for parallel spins ($e^-$ : $\boldsymbol{Q} = 1$, $e^+$ : $\boldsymbol{P}_T = 1$) and for perpendicular spins. The maximum of the intensity lies on the bisector of the angle $\omega t$ between the two spins. Thus the "figure of eight" moves with angular frequency $\omega/2$. For a fixed detector pair at azimuthal angle $\psi$ the time dependence is still given by the angular frequency $\omega$ due to the two symmetric lobes of the "figure of eight". From [11, 23].

the nature of the two unobserved neutral particles, time reversal invariance (disregarding the negligible final state interactions) requires $P_{T_2} = 0$.

The measurement of $\boldsymbol{P}_T$ as a function of energy yields a determination of the parameters $\eta$, $\eta''$, $\alpha'/A$ and $\beta'/A$ (see Eqs. (6.16), (6.17), (6.23d) and (6.23h)). $\eta$ is of special interest. $\eta$, together with the Michel parameter $\varrho$, determines the shape of the (isotropic) positron energy spectrum. However, it is difficult to deduce its value from a spectrum measurement, as its influence is suppressed by a factor $x_0 \approx 10^{-2}$. On the other hand, a precise value is needed for a precise determination of $\varrho$, as $\eta$ and $\varrho$ are statistically highly correlated. In (6.14c) for $P_{T_1}$, $\eta$ arises without a suppression factor. lt is interesting to note that $P_{T_1}$ does not vanish in the Standard Model interaction, as can be seen from (6.9), and it may take sizeable values ($|P_{T_1}| \leq 1/3$) for positron energies of a few MeV.

The experiment was performed with basically the same setup used for measuring the longitudinal polarization. It also uses a comparison with the spin polarized electrons in a ferromagnetic foil from annihilation in flight $e^+e^- \to \gamma\gamma$. lt is based on the fact that the photons from the annihilation of a relativistic, transversely polarized positron electron pair are preferentially emitted in the plane defined by the particle line-of-flight $\boldsymbol{k}_{e^+}$ and the bisector $\boldsymbol{b}$ between the (transverse) polarization directions $\boldsymbol{p}_T$ and $\boldsymbol{p}_{e^-}$ (see Figure 6.3).

The results of a general, unrestricted analysis of the data are an improved value for $\eta = (11 \pm 85) \times 10^{-3}$ and the first results for $\eta'' = (48 \pm 125) \times 10^{-3}$ and the T-violating parameters $\alpha'/A = (-47 \pm 52) \times 10^{-3}$ and $\beta'/A = (17 \pm 18) \times 10^{-3}$ [13].

An improved experiment, where all the major parts of the previous experiment have been replaced by newly designed equipment to increase the event rate and reduce the systematic errors, has been described in detail elsewhere [24]. The four NaI detectors were replaced by an array of 127 BGO detectors (see Figure 6.4). A longitudinally polarized $\mu^+$ beam ($P_\mu^b = 91\,\%$) enters a beryllium stop target with bunches every 19.75 ns. The polarization $P_\mu(t)$ of the stopped muons precesses in a homogeneous magnetic field ($B = 373.6 \pm 0.4$ mT) with the same angular frequency $\omega$ as the accelerator radio frequency. This ensures that $P_\mu(t) \parallel P_\mu^b$ for

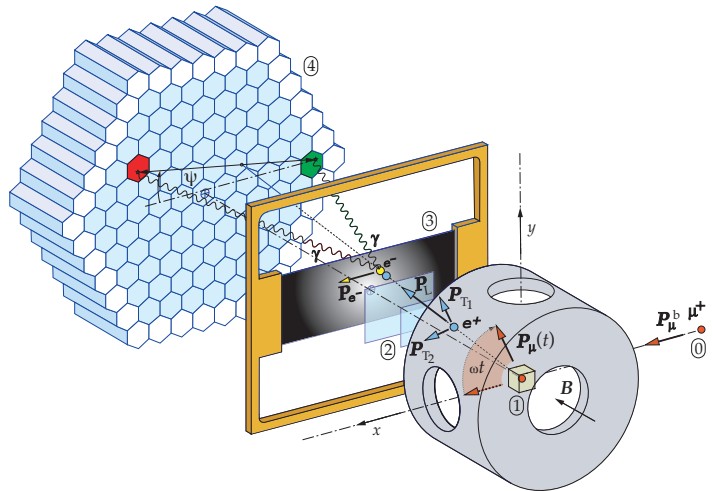

Figure 6.4: Schematic view of the experimental setup for the measurement of $\boldsymbol{P}_\mathrm{T}$. **0:** Burst of polarized muons (angular frequency $\omega$, polarization $\boldsymbol{P}_\mu^\mathrm{b}$). **1:** Be stop target and precession field $\boldsymbol{B}$. **2:** Two plastic scintillation counters selecting decay positrons **3:** Magnetized Vacoflux 50$^\mathrm{TM}$ foil serving as a polarization analyzer. **4:** Array of 127 BGO scintillators to detect the two $\gamma$'s from $e^+$ annihilation-in-flight. From [25].

each newly arriving $\mu^+$ bunch. Because of the burst width of 3.9 ns (FWHM) the polarization $P_\mu(0)$ of the stopped $\mu^+$ is reduced to $(82 \pm 2)\%$. A system of drift chambers (not shown) and two thin plastic scintillator counters $T_0$ and $T_1$ select decay $e^+$'s emitted in the direction of $\boldsymbol{B}$. A 1-mm-thick magnetized Vacoflux 50$^\mathrm{TM}$ foil (49 % Fe, 49 % Co, 2 %V) in the central region with its polarized $e^-$ ($P_{e^-} = 7.2\,\%$) serves as polarization analyzer. The two $\gamma$'s from $e^+$ annihilation-in-flight with the polarized $e^-$ are selected by an array of 91 interior $\mathrm{Bi}_4\mathrm{Ge}_3\mathrm{O}_{12}$ (BGO) crystals with plastic veto counters in front of them to reject charged particles. The outer layer of 36 BGOs assists in an efficient collection of the deposited energy. Valid events are selected by using the correlation between the $\gamma$ energies and their opening angle. The intensity distribution of the two $\gamma$'s has roughly the shape of the figure eight with a maximum in the direction of the bisector of $\boldsymbol{P}_\mathrm{T}(t)$ and the $e^-$ polarization $\boldsymbol{P}_{e^-}$ [11, 23] (see Figure 6.3). The precession of $\boldsymbol{P}_\mu(t)$ implies a precession of $\boldsymbol{P}_\mathrm{T}(t)$, while $\boldsymbol{P}_{e^-}$ remains constant in time. Thus the intensity distribution of the $\gamma$'s also precesses with frequency $\omega$. For any given pair $ij$ of BGO detectors we ideally expect a signal for the normalized annihilation rate $N_{ij}(t)$ in the form

$$N_{ij}(t) = 1 + a_{ij}\cos(\omega t + \delta_0) + b_{ij}\sin(\omega t + \delta_0), \tag{6.45}$$

where $t$ denotes the time the $e^+$ traverses counter $T_0$ and $\delta_0$ an instrumental phase common to all time spectra. The events are contained in a time window of 39.5 ns total width, corresponding to two periods of the accelerator RF. The Fourier coefficients $a_{ij}$ and $b_{ij}$ contain the complete information of the transverse positron polarization. The analyzing power for annihilation in flight is large in most of the kinematic regions of the experiment. Figure 6.5 shows, as an example, the contour lines for the transverse analyzing power $A_x$ (in %) as a function of the sum $u = (E_3 + E_4)/m_e$ and the difference $v = (E_3 - E_4)/m_e$ of the normalized photon energies $E_3$ and $E_4$.

Due to the finite acceptance solid angle for events, the rate of ANN events also varies with the frequency $\omega$ because of a small muon spin rotation ($\mu$SR) decay asymmetry modulated by the precessing $\boldsymbol{P}_\mu(t)$. By adding or subtracting the Fourier coefficients of appropriate pairs

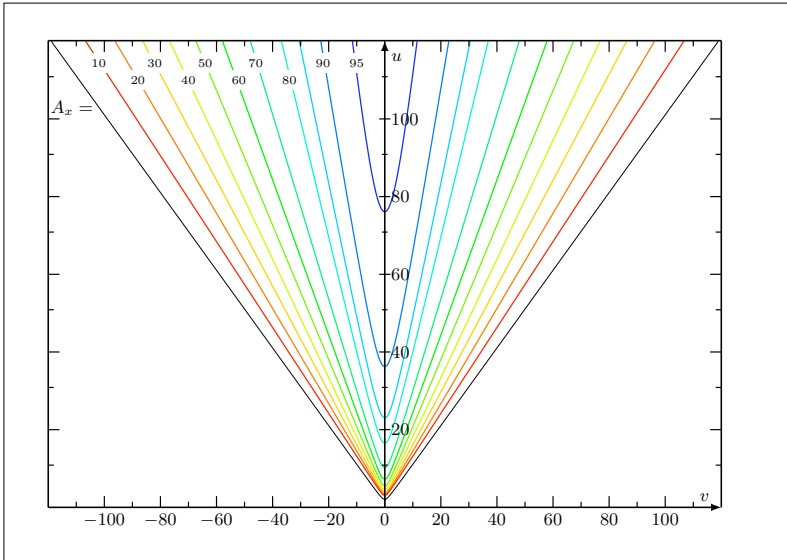

Figure 6.5: Contour lines for the transverse analyzing power $A_x$ (in %) as a function of the sum $u = (E_3 + E_4)/m_e$ and the difference $v = (E_3 - E_4)/m_e$ of the normalized photon energies $E_3$ and $E_4$. The outermost line is the kinematic boundary. From [11].

$ij$ and $i'j'$, it was possible to derive either the $\mu$SR - or the $P_T$ signal, respectively. The $\mu$SR signal is essential for the experiment, as it allows the decomposition of the vector $\boldsymbol{P}_T$ into its components $(P_{T_1}, P_{T_2})$, since $\boldsymbol{P}_{T_1}$ lies in the plane of $\boldsymbol{k}_{e^+}$ and $\boldsymbol{P}_\mu(t)$ and $\boldsymbol{P}_{T_2}$ perpendicular to that plane (see Figure 6.1).

Table 6.1 shows the results of the general and of a restricted analysis [25]. The average polarization components $\langle P_{T_1} \rangle$ and $\langle P_{T_2} \rangle$ have been calculated from the values of $\eta$, $\eta''$, and $\alpha'/A$, $\beta'/A$, respectively. Based on the most general 4-fermion contact interaction ("general analysis") the parameter $\eta$ is given by [12]

$$\eta = \frac{1}{2} Re \left\{ g_{LL}^V g_{RR}^{S*} + g_{RR}^V g_{LL}^{S*} + g_{LR}^V \left( g_{RL}^{S*} + 6 g_{RL}^{T*} \right) + g_{RL}^V \left( g_{LR}^{S*} + 6 g_{LR}^{T*} \right) \right\}. \tag{6.46}$$

With $g_{LL}^V \approx 1$, and all other $g_{\varepsilon\mu}^\gamma \approx 0$ [7], one can simplify (6.46) considerably by neglecting all terms quadratic in non-standard couplings. This amounts to assuming one additional coupling beyond $V - A$. Then only two independent parameters remain ("restricted analysis"):

$$\eta = \tfrac{1}{2} Re\{g_{RR}^S\}, \quad \beta'/A = -\tfrac{1}{4} Im\{g_{RR}^S\}. \tag{6.47}$$

Here, $g_{RR}^S$ is a scalar coupling with right-handed $\mu$ and $e$.

The Fermi coupling constant $G_F$ is generally derived assuming an exclusive $V - A$ interaction, which amounts to setting $\eta = 0$. However, $G_F$ depends on $\eta$ [2, 12]:

$$G_F \approx G_F^{V-A} \cdot \left( 1 - 2\eta \frac{m_e}{m_\mu} \right), \tag{6.48}$$

where $m_e/m_\mu$ is the mass ratio of electron and muon. Taking $\eta$ into account increases the relative error $\Delta G_F/G_F$ from $9 \times 10^{-6}$ to $360 \times 10^{-6}$ (general analysis) resp. to $68 \times 10^{-6}$ (restricted analysis).

Note that the results on $\alpha'/A$, $\beta'/A$ (and deduced from these, $\langle P_{T_2} \rangle$ and Im $\{g_{RR}^S\}$) are the only experimental data sensitive to the violation of time reversal invariance (TRI) for a purely

Table 6.1: $V - A$ values and experimental results. All values, except $\chi^2$/d.o.f., in units of $10^{-3}$. The correlation coefficients $\rho_{ij}$ are all compatible with zero except the two coefficients listed. The errors are statistical and systematic.

| | $V-A$ | General analysis | Restricted analysis |
|---|---|---|---|
| $\eta$ | 0 | $71 \pm 37 \quad \pm 5$ | $-2.1 \pm 7.0 \pm 1.0$ |
| $\eta''$ | 0 | $105 \pm 52 \quad \pm 6$ | $\equiv -\eta$ |
| $\alpha'/A$ | 0 | $-3.4 \pm 21.3 \pm 4.9$ | $\equiv 0$ |
| $\beta'/A$ | 0 | $-0.5 \pm 7.8 \quad \pm 1.8$ | $-1.3 \pm 3.5 \pm 0.6$ |
| $\rho_{\eta\eta''}$ | | 946 | $-$ |
| $\rho_{\alpha'\beta'}$ | | -893 | $-$ |
| $\chi^2$/d.o.f. | | 46.2/33 | 50.3/35 |
| Re $\{g_{RR}^S\}$ | 0 | $-$ | $-4.2 \pm 14.0 \pm 2.0$ |
| Im $\{g_{RR}^S\}$ | 0 | $-$ | $5.2 \pm 14.0 \pm 2.4$ |
| $\langle P_{T_1}\rangle$ | -3 | $6.3 \pm 7.7 \quad \pm 3.4$ | |
| $\langle P_{T_2}\rangle$ | 0 | $-3.7 \pm 7.7 \quad \pm 3.4$ | |

leptonic *reaction*. In contrast to the violation of TRI in the neutral kaon system [26], a $T$-odd observable in muon decay would be due to an interference between two couplings with different phase angles and thus be an unambiguous signal of new physics beyond the Standard Model.

### 6.5.3 Electron Decay Asymmetry

The measurement of the electron decay asymmetry, $\mathcal{A}(x)$, from polarized muons [27], determines how strongly the chiral components $(L, R)$ of the muon take part in the interaction. This has been used to search for right-handed currents and other muon decay modes outside the Standard Model.

If the combination

$$\tfrac{1}{18}(9 + 3\xi - 16 \cdot \xi \cdot \delta) = \tfrac{1}{4}|g_{RR}^S|^2 + \tfrac{1}{4}|g_{LR}^S|^2 + |g_{RR}^V|^2 + |g_{LR}^V|^2 + 3|g_{LR}^T|^2$$
$$\equiv Q_{RR} + Q_{LR} \equiv Q_R^\mu \tag{6.49}$$

has a value different from zero, then a coupling to the right-handed component of the muon has to exist, i.e. at least one $g_{\varepsilon R}^\gamma \neq 0$. Conversely, if $Q_R^\mu = 0$, then the coupling acts exclusively on the left-handed muon.

The distribution of the flight direction of the positrons (electrons) is given by (6.5) with $\boldsymbol{P}_e = 0$ as

$$\frac{\mathrm{d}^2\Gamma}{\mathrm{d}x\,\mathrm{d}\cos\vartheta} \equiv w(x, \vartheta) \sim \left\{ F_{\mathrm{IS}}(x) \pm P_\mu \cos\vartheta F_{\mathrm{AS}}(x) \right\} . \tag{6.50}$$

This depends on the reduced energy, $x$, the angle $\vartheta$ between the muon polarization and the positron momentum as chosen by the detector, and on the degree of polarization $P_\mu > 0$. The asymmetry

$$\mathcal{A}(x) \equiv \frac{w(x, 0) - w(x, \pi)}{w(x, 0) + w(x, \pi)} = P_\mu \cdot \frac{F_{\mathrm{IS}}(x)}{F_{\mathrm{AS}}(x)} \tag{6.51}$$

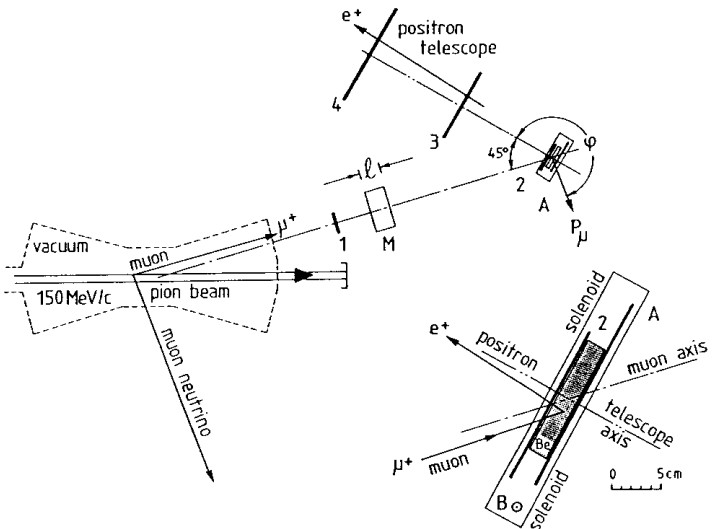

Figure 6.6: Muon Spin Rotation apparatus used to measure the integral asymmetry of the $e^+$ direction distribution following the decay of highly polarized muons. A parallel beam of monoenergetic (150 MeV/$c$) pions decays in flight in vacuum. Muons with energies within a well-determined interval are selected to stop in a beryllium plate, Be, employing a moderator of length $\ell$. The original orientation of the muon polarization vector $P_\mu$ is thus defined. A rectangular solenoid produces a vertical magnetic field $B = 3$ mT causing the polarization of the stopped muons to precess in the horizontal plane. This gives rise to a sinusoidal modulation of the exponential decrease of the positron rate. The amplitude of the modulation ($\approx 1/3$) is proportional to the quantity desired, $P_\mu \xi$. From [27].

depends on the parameters $\varrho$, $\eta$, $\xi$ and $\xi\delta$ (see Eqs. (6.13a), (6.13b), (6.14a) and (6.14b)).

The distributions of the flight directions of the positrons (electrons) as seen by an apparatus that is equally sensitive to positrons of all energies is given by

$$\frac{d\Gamma}{d\cos\vartheta} \sim \int_{x_0}^{1} dx \cdot \sqrt{x^2 - x_0^2} \cdot F_{IS}(x) \pm P_\mu \cos\vartheta \cdot \int_{x_0}^{1} dx \cdot \sqrt{x^2 - x_0^2} \cdot F_{AS}(x)$$
$$\sim (1 \pm \mathcal{A}' \cdot \cos\vartheta). \tag{6.52}$$

The integral asymmetry, $\mathcal{A}'$, is proportional to $P_\mu \cdot \xi$ and depends on $\eta$ in first order and on $\delta$ in second order of $x_0$. Neglecting $x_0$ ($x_0 = 0$) one obtains

$$\mathcal{A}' = \tfrac{1}{3} \cdot P_\mu \cdot \xi. \tag{6.53}$$

This allows the determination of $\xi$ from an experiment using muons of known polarization. In the analysis, the knowledge of the values of other muon decay parameters is unimportant.

Muon beams produced from pions decaying in flight in vacuum avoid Coulomb multiple scattering. The muon spin lies in the plane of the laboratory line of flight of the original pion, $k_\pi$, and and its decay muon, $k_\mu$. It points inwards (towards $k_\pi$) for $\mu^+$ and outwards for $\mu^-$ (see Figure 6.6). The transverse and longitudinal muon spin components, $\zeta_T$ and $\zeta_L$ with

respect to the muon's laboratory line-of-flight are simply given by

$$\zeta_T = \frac{\sin\vartheta_\mu}{\sin\Theta_\mu}, \qquad\qquad \zeta_L = \mp\sqrt{1-\zeta_T^2}, \qquad\qquad (6.54)$$

where the upper (lower) sign applies for the muon emitted with smaller (larger) momentum for the given angle of emission $\vartheta_\mu$, and where

$$\vartheta_\mu = \text{laboratory angle between } \boldsymbol{k}_\pi \text{ and } \boldsymbol{k}_\mu,$$
$$\Theta_\mu = \text{maximum laboratory angle by kinematics (Jacobian peak angle)},$$
$$\sin\Theta_\mu = \left(m_\pi^2 + m_\mu^2\right)/(2m_\pi k_\pi),$$
$$k_\pi = \text{pion beam momentum}.$$

The selection of a small slice of muon energy in the laboratory in the vicinity of the Jacobian peak corresponds to a choice of a small range of neutrino directions and thus of a degree of polarization $P_\mu = G \cdot P_{\nu_\mu}$. The geometrical factor $G$, which also has been studied experimentally [28], is close to one ($> 0.99$), and it is known with an uncertainty of $< 10^{-3}$ [27].

To measure the decay asymmetry, the muons are stopped in a metal (Be, Al) immersed in a transverse magnetic field where the spins precess. Detectors track the muon and the decay positron momenta. The positron intensity has a time modulation corresponding to the decay asymmetry. It is fortunate that there are substances (Al, Cu, Ag, Au, bromoform) that barely influence the spin direction of muons inside them. The disappearance of muon polarization during slowing down [21, 29] and thermalisation [30], i.e. at earlier times compared to the muon precession time, mimics a smaller $\mathcal{A}_{\text{exp}}$. Depolarization at later times is seen in the data [31,32]. It can be accounted for by extrapolating the precession signal amplitude to time zero. The determination of the extrapolating-function parameters in the same experiment generally considerably reduces the statistical significance of the data due to their strong correlation with the signal. The relaxation time in pure metals at room temperature is often conveniently large compared to the muon lifetime.

Positron detectors with low energy thresholds are used for the measurement of $P_\mu\xi$. The result obtained from this experiment is [27]

$$P_\mu^\pi\xi = (1002.7 \pm 7.9_{\text{stat.}} \pm 3.0_{\text{syst.}}) \times 10^{-3}. \qquad\qquad (6.55)$$

As $\xi$ is not limited close to the measured value of $P_\mu\xi$, we cannot draw any specific conclusion on $P_\mu$ and $\xi$ separately. In fact, $-3 \le \xi \le +3$. To isolate $\xi$ from $P_\mu\xi$, one has to deduce $P_\mu$ from the measurement of $P_\mu\xi\delta/\varrho$ of [32].

## 6.6 Results for $\tau$-lepton and neutrino physics

For muon decay, we have shown that a hamiltonian with parity-odd and -even terms is not well suited for the description of a fully parity-violating interaction. Thus we have extended the concept of the *chiral* hamiltonian to leptonic $\tau$ decays [33]. Assuming universality for leptonic $\tau$ decays sensitivities for the different $\tau$ decay constants can be derived.

For the complete determination of the interaction in muon decay, it was essential to have experimental proof that the helicity of left-handed $\nu_\mu$ is equal to $-1$. Previous measurements had yielded $h_{\bar{\nu}_\mu} = (+990 \pm 160) \times 10^{-3}$ [34] and $h_{\nu_\mu} = (-1060 \pm 110) \times 10^{-3}$ [35]. It was then realized that the measurement of $P_\mu\xi\delta/\varrho$ in muon decay by Carr et al. [36] not only yields a new lower limit for a possible right-handed $W_R$ boson, but is also suited to derive a vastly improved limit for the helicity of the $\nu_\mu$ [37]:

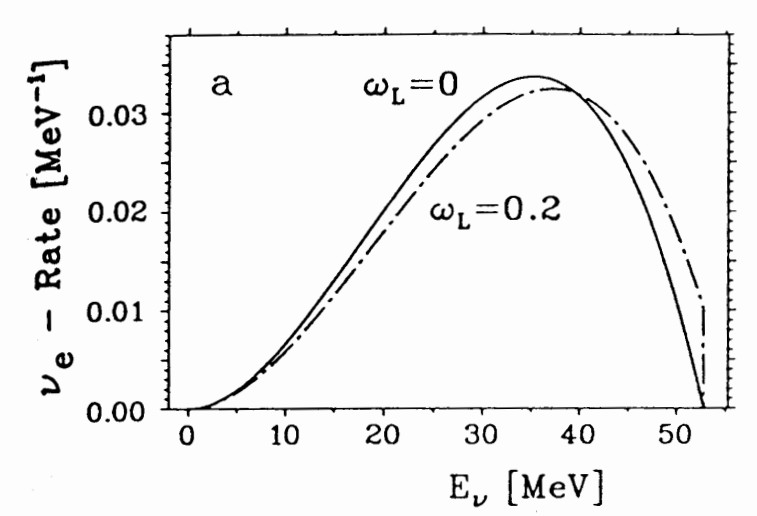

Figure 6.7: Normalized energy distributions of left-handed $\nu_e$ from the decay of unpolarized $\mu^+$. The spectrum shape parameter $\omega_L$ is the analog of the Michel parameter $\varrho$ of the $e^+$. For a pure $V-A$ interaction $\omega_L$ is equal to zero. From [38].

The normalized positron rate $\mathrm{d}^2\Gamma/\mathrm{d}x\,\mathrm{d}\cos\vartheta$ at the spectrum end point can be written as

$$\frac{\mathrm{d}^2\Gamma}{\mathrm{d}x\,\mathrm{d}\cos\vartheta} = (1 + P_\mu \cdot (\xi\delta/\varrho) \cdot \cos\vartheta). \tag{6.56}$$

It is obvious that the factor $|P_\mu \xi\delta/\varrho| \leq 1$, since the rate cannot be negative. $P_\mu$ is the polarization of the muon from the decay $\pi^+ \to \mu^+\nu_\mu$ and independent of the muon decay constant. Therefore we find

$$|P_\mu| \leq 1 \quad \text{and} \quad |\xi\delta/\varrho| \leq 1. \tag{6.57}$$

On the other hand, from the measurement one gets a lower limit for the product [36]

$$P_\mu \xi\delta/\varrho > 995.9 \times 10^{-3} \quad (90\%\text{CL}). \tag{6.58}$$

Since $P_\mu = -h_{\nu_\mu}$ we derive a lower limit for $|h_{\nu_\mu}|$ [37]:

$$|P_\mu| = |h_{\nu_\mu}| > 995.9 \times 10^{-3} \quad (90\%\text{CL}). \tag{6.59}$$

It has also been realized that experiments that detect the $\nu_e$ from the decay of unpolarized $\mu^+$ by the reaction $^{12}\mathrm{C}(\nu_e, e^-)^{12}\mathrm{N}(\text{g.s.})$ not only determine the neutrino absorption cross section but also measure the $\nu_e$ energy spectrum [38]. The energy spectrum can be described by the spectrum shape parameters $\omega_L$ and $\eta_L$ for left-handed and $\omega_R$ and $\eta_R$ for right-handed $\nu_e$. In contrast to the energy spectrum of the electrons it allows a new null-test of the standard model [38]. The right-handed $\nu_e$ cannot be detected as they are sterile in matter. For the energy spectrum of the left-handed $\nu_e$ one obtains

$$\frac{\mathrm{d}\Gamma_L}{\mathrm{d}y} = \frac{m_\mu^5 G_F^2}{16\pi^3} \cdot Q_L^{\nu_e} \cdot \{F_1(y) + \omega_L \cdot F_2(y) + \eta_L x_0 F_3(y)\}. \tag{6.60}$$

Here, $\mathrm{d}\Gamma/\mathrm{d}y$ is the probability of a left-handed $\nu_e$ to be emitted with the reduced energy $y = 2E_\nu/m_\mu$. The functions $F_1(y)$, $F_2(y)$ and $F_3(y)$ are given in [38]. The probability $Q_L^{\nu_e}$ of

the $\nu_e$ to be left-handed, the spectral shape parameter $\omega_L$ and the low energy parameter $\eta_L$ are

$$Q_L^{\nu_e} = \tfrac{1}{4}|g_{RL}^S|^2 + \tfrac{1}{4}|g_{RR}^S|^2 + |g_{LL}^V|^2 + |g_{LR}^V|^2 + 3|g_{RL}^T|^2 = \tfrac{1}{2}\left(1 - P_{\nu_e}\right), \qquad (6.61)$$

$$\omega_L = \tfrac{3}{4}\frac{\left\{|g_{RR}^S|^2 + 4|g_{LR}^V|^2 + |g_{RL}^S + 2g_{RL}^T|^2\right\}}{\left\{|g_{RL}^S|^2 + |g_{RR}^S|^2 + 4|g_{LL}^V|^2 + 4|g_{LR}^V|^2 + 12|g_{RL}^T|^2\right\}}, \qquad (6.62)$$

$$\eta_L = 2\frac{Re\left\{g_{LL}^V g_{RR}^{S*} + g_{LR}^V\left(g_{RL}^{S*} + 6g_{RL}^{T*}\right)\right\}}{\left\{|g_{RL}^S|^2 + |g_{RR}^S|^2 + 4|g_{LL}^V|^2 + 4|g_{LR}^V|^2 + 12|g_{RL}^T|^2\right\}}, \qquad (6.63)$$

where $P_{\nu_e}$ denotes the longitudinal polarization of the $\nu_e$. Figure 6.7 , as an example, shows the normalized energy distributions for the $V-A$ prediction $\omega_L^{V-A} = 0$ and for $\omega_L = 0.2$. A value $\omega_L > 0$ results in events at the spectrum end where none are expected for the $V-A$ interaction.

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
