# Peer review of "Muon Decay"

_SciPost Physics Proceedings, doi:SciPost Phys. Proc. 5, 006 (2021)_

## Round 1 · Referee Report · Anonymous (Referee 1) · 2021-6-22

Report

Please see PDF

Attachment

  • validity: -
  • significance: -
  • originality: -
  • clarity: -
  • formatting: -
  • grammar: -

Author:  Wulf Fetscher  on 2021-07-04  [id 1543]

(in reply to Report 1 on 2021-06-22)
Category:
reply to objection

I appreciate the thorough report of the referee. Let me begin with comments to the content:

26) In this approximation the interaction is local, 27) and following: replaced "point interaction" by "contact interaction", 28) Yes, it refers to Fierz identities, 31) I agree, the physics does not depend on the basis. On the other hand, it seemed reasonable to include lepton number conservation which implied the change from charge retention to charge changing form. 37-38) This statement is correct; there really are four coupling constants, and all leptons are left-handed. 40) Replaced "handedness" by "chirality"; 40) who in turn use -> which in turn uses, 41) I believe it suffices to continue with the matrix element where the relevant coupling constants are defined. 42) G_F; since F stands for Fermi, it is not unobservable and it should be roman and not italic. 43-49) The chiralities are indeed fixed by \gamma: For \gamma=V the chirality of the outgoing particle is equal to the chirality of the incoming particle; for scalar and tensor interaction, the chirality changes. 50) I use „definitely“ because of widespread use of the term right-and left-handed both for the spinors and for the helicities. 53) A left-handed spinor transforms under a Lorentz boost always to a left-handed spinor part of (complex ) C2. 57) I prefer to introduce reduced energy already here and define air shortly after because I need to define W_e\mu first. 59) In order not to define not too many observables, I prefer to start with a general formula which displays the angular and energy distribution. Thus the detailed definition of P_T1, P_T2 and P_L strongly relies on Figure 1. 65) At this point it suffices to mention that the decay distribution depends on these four decay parameters. They will be discussed in detail with the relevant measurements. 68) The SM is only one of the ten interactions treated, so masses could be different from zero. 71) x is the reduced energy, while x_1, x_2 and x_3 are vectors in space. 72) For the Standard-Model coupling -> Within the Standard Model. 79) CP-violation would follow for CPT invariance. 84) Here, only effects of the normalized decay distributions are discussed. 111) about the decay interaction - > about the interaction 113) yielding the V-A value of 3/4-> yielding the value 3/4 as predicted by V-A. 116) True 118) decay interaction -> interaction 125) and the others -> while the others 126) Yes, they are! 129) muon and electron observables 133) Yes, the muon lifetime is also needed. 179) The transverse…

Comments to the style: I have added commas and periods to equations, as suggested by the referee. On the other hand I will keep my way of citing equations without the Eq. in front of the equation number, since this is a style question and is used by the authors of the other PSI papers.

---

## Round 1 · Referee Report · Adrian Signer (Referee 2) · 2021-7-5

Report

We (the editors Cy Hoffman, Klaus Kirch, Adrian Signer) had the
opportunity to review an earlier draft of the article and were in
communication with the authors before the submission. All our comments
and suggestions have been taken into account. Hence, we think the
paper can now be published in the current form.

---

## Round 2 · Referee Report · Anonymous (Referee 1) · 2021-7-6

Report
Eq. (6.1) and the discussion around it should still be improved. The titel of the section is Hamiltonian, but no Hamiltonian is given. The Gamma (capital letter) are not defined but called interactions, which is misleading as they are Dirac gamma matrices. There should also be no ident after this equation.

---

## Round 2 · List of Changes

1) I Added the definition of inverse muon decay.
2) Now the coordinate system in Fig. 6.1 agrees with the definition of the transverse polarization.
3) I added a sentence to reference Fig. 6.7.

---

## Round 3 · Author Response

The detailed form of the Gamma^\gamma matrices is not relevant for the rest of this paper, since the important physics is contained in the ten complex coupling constants.

---

## Round 3 · List of Changes

I have changed the section title: 6.2 Hamiltonian -> General Matrix Element;
It is now noted that the \Gamma^\gamma are either Dirac or Pauli matrices.
\noindent is used after Eq. (6.1)

---

## Editorial Decision

published